# Fine-Grained Reward Modeling in LLMs: An RL, PRM, and Memory-Augmented Approach for Advanced Reasoning

**Jiaxiang Liu**
ID: 2024211331
jiaxiang24@mails.tsinghua.edu.cn

**Wanlan Ren**
ID: 2023210455
rwl23@mails.tsinghua.edu.cn

## 1  Background

Traditional alignment methods for large language models (LLMs) such as GPT-3 and GPT-4 primarily use Reinforcement Learning from Human Feedback (RLHF) to optimize behavior. These methods, however, often rely on broad and simplistic reward categories like "usefulness" and "toxicity." While effective for basic alignment, this limited framework fails to capture the complexity needed to align LLMs with nuanced human preferences[1], especially in sophisticated applications such as dialogue systems, customer service automation, and multi-step problem-solving. Additionally, existing methods use Process Reward Models (PRMs) to evaluate intermediate steps in the reasoning process, but they lack mechanisms to dynamically adapt or utilize past information efficiently.

To address these limitations, we propose three key enhancements: **diversifying reward metrics**, introducing **another LLM for automated feedback evaluation**, and incorporating a **working memory mechanism**. We expand the reward model to include metrics like "contextual coherence," "logical consistency," and "goal alignment." This approach enhances adaptability and reduces manual annotation costs, enabling real-time, task-specific evaluations. The memory mechanism dynamically updates to manage information flow effectively during multi-step reasoning.

## 2  Definition

We aim to develop a model that leverages reinforcement learning (RL), a process reward model (PRM), and a memory mechanism to dynamically optimize the inference process of large language models (LLMs). The problem is defined as follows:

Let $M$ denote the large language model, and $T$ represent a task requiring multiple reasoning steps $\{s_1, s_2, \ldots, s_n\}$. An RL agent's policy $\pi_\theta$ guides the action $a_t$ at each step $s_t$, influenced by evaluations from the PRM and the memory state $WM_t$.

The model function is expressed as:

$$M(X) = f(X, WM_t)$$

where $WM_t$ is the memory state at time $t$, updated based on PRM evaluations and RL feedback.

The reward function is:

$$R(s_t, a_t) = \sum_{i=1}^{k} \beta_i \cdot PRM_i(s_t, a_t) + \gamma \cdot g(WM_t),$$

where $PRM_i(s_t, a_t)$ denotes the score from the $i$-th PRM module evaluating criteria such as logic, coherence, and goal alignment; $\beta_i$ are the weights assigned to each PRM component; $g(WM_t)$ represents the memory utility at time $t$; and $\gamma$ adjusts its influence within the reward function.

Preprint. Under review.

# 3   Related Work

Several studies have investigated the integration of RL and PRM in the context of large language models (LLMs).

Ziegler et al. (2019) [2] introduced Reinforcement Learning from Human Feedback (RLHF), where an RL agent optimizes LLM behavior based on human-provided feedback. Although effective, this approach relies on basic reward functions like "usefulness" and "toxicity," lacking the granularity needed for more complex tasks.

OpenAI employs a Process Reward Model (PRM) to evaluate each reasoning step [3], ensuring intermediate step consistency. However, this model does not incorporate memory mechanisms for recalling prior states, limiting its effectiveness in handling multi-step reasoning.

Memory-augmented networks have also been explored. Graves et al. (2016) [4] introduced Differentiable Neural Computers (DNCs), incorporating memory modules to efficiently store and recall information. However, these models lack the integration of RL for dynamic, task-specific adaptation. Similarly, Weston et al. (2014) [5] and Santoro et al. (2018) [6] explored Memory Networks and Relational Memory Networks, respectively, but did not integrate reinforcement learning or fine-grained reward modeling.

Our approach builds on these works by combining RL, PRM for step-by-step evaluation, and a memory mechanism for dynamic adaptation, addressing the limitations in handling complex multi-step tasks.

# 4   Proposed Method

Our approach addresses the limitations of traditional RLHF by enhancing reward modeling and integrating memory mechanisms to support complex reasoning tasks. We introduce fine-grained reward metrics such as "contextual coherence" and "logical consistency," allowing for more comprehensive and adaptive evaluation beyond simple "usefulness" and "toxicity" criteria. By utilizing another LLM as an automated evaluator, our system reduces manual labeling costs and adapts dynamically to diverse tasks, improving model performance across various applications.

The incorporation of a working memory module further enhances our method. This module, guided by a gating mechanism, dynamically manages the storage, recall, or update of information based on task complexity, allowing effective multi-step reasoning and improving inference accuracy and efficiency.

To validate our approach, we use the following datasets: **SQuAD 2.0** for testing question-answering capabilities, **NarrativeQA** for assessing coherence in long-form narratives, and **bAbI Tasks** for evaluating multi-step and memory-dependent reasoning.

Our method will be compared against baselines such as:

- **Traditional RLHF**: Uses basic criteria and manual labels.
- **PRM-Only Framework**: Evaluates steps without memory integration.
- **Memory-Augmented Models**: Such as Differentiable Neural Computers (DNCs) without RL integration.

To implement our method, the PRM modules will evaluate multiple dimensions such as logic, coherence, and goal alignment. The memory module will be designed to store intermediate results, guided by a gating mechanism to determine when information should be recalled or updated. An RL agent, trained using Proximal Policy Optimization (PPO), will utilize feedback from both PRM and the memory module to optimize its actions at each step, maximizing cumulative rewards over the reasoning sequence. Additionally, we introduce an iterative self-reward mechanism, where the evaluation and feedback cycle is repeated over multiple rounds, refining the RL strategy through experience replay, allowing the model to learn from past memory usage and PRM evaluations, thus ensuring more robust and adaptive learning.

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
