# OpenReview forum: "【Proposal】Fine-Grained Reward Modeling in LLMs: An RL, PRM, and Memory-Augmented Approach for Advanced Reasoning"
_tsinghua.edu.cn/THU/2024/Fall/AML — THU 2024 Fall AML Submission_

### Official Review · ~Yinuo_Li1 · 2024-11-06
**Good idea and approach**

**Rating:** 9
**Confidence:** 3

**Review:**

This proposal clearly showed their research topic and their approach method, the problem is well defined.

They mainly proposed three goals to achieve: 1. using diverse reward metrics rather than just use broad and simplistic reward categories like “usefulness” and “toxicity”; 2. using a LLM for automatic reward feedback; 3.using working memory for better understanding to previous information.

Their three main components are all valuable and useful in alignment for LLM, diverse reward metric allows more evaluating aspects, therefore more information can be captured by the feedbacks; automatic feedback evaluating produce high efficiency and working memory allows the system to keep important previous information, therefore allows longer reasoning steps.

However, I am a bit not sure about the performance of the automated evaluator compared with human evaluator and it may take even more efforts to make sure this LLM is neutral and aligned with human expectation.

---

### Official Review · ~Xun_Wang10 · 2024-11-10
**Review for "Fine-Grained Reward Modeling in LLMs: An RL, PRM, and Memory-Augmented Approach for Advanced Reasoning"**

**Rating:** 10
**Confidence:** 4

**Review:**

This study proposes an advanced framework for large language model (LLM) alignment, integrating Reinforcement Learning (RL), a Process Reward Model (PRM), and a dynamic memory mechanism, to address the limitation of traditional RLHF approaches. This architecture offers improved scalability and accuracy, reducing reliance on manual labeling and enhancing inference quality.

Strength: The proposal provides a detailed background on the problem, clearly defines the proposed hybrid model, and analyzes the limitations of existing work. It describes the proposed method clearly and outlines the means for evaluating its effectiveness.

Weakness: Adding an abstract section would improve the proposal.

---

### Official Review · ~Kehan_Zheng1 · 2024-11-11
**Excellent Proposal**

**Rating:** 10
**Confidence:** 4

**Review:**

This proposal is well-structured, offering a clear background, problem definition, and innovative approach to fine-grained reward modeling in LLMs. The method, which combines reinforcement learning, process reward models, and memory mechanisms, aims to solve the limitations of traditional RLHF, especially in complex reasoning tasks. It can be better if you can make some explanations on the use of LLM evaluators, since as we know, LLMs can sometimes be biased in judgement.

---

### Official Review · ~Tianhai_Liang1 · 2024-11-11
**Great Proposal**

**Rating:** 10
**Confidence:** 4

**Review:**

Existing Large Language Models (LLMs) using Reinforcement Learning from Human Feedback (RLHF) rely on simplistic reward metrics, which are inadequate for complex tasks like dialogue systems and multi-step reasoning. Current Process Reward Models (PRMs) lack dynamic adaptation and memory utilization. The proposal offers a novel approach to improving LLM performance on complex tasks that combines Fine-Grained Reward Metrics, Automated Feedback, and Memory Mechanism. The model is trained using Proximal Policy Optimization (PPO) with feedback from PRMs and the memory module. The reward function combines PRM evaluations and memory utility.

The proposal clearly defines the research problem, presents a well-structured approach, and offers detailed explanations of each component of the hybrid model. The integration of diverse reward metrics and automated feedback evaluation is well justified, and the method for assessing its performance using multiple datasets is clearly outlined.

However, the proposal would be strengthened by including more concrete experimental plans and benchmarks for evaluating the proposed memory mechanism's effectiveness.

---

### Official Review · ~Yida_Lu1 · 2024-11-11
**Good idea and well-structured proposal**

**Rating:** 9
**Confidence:** 4

**Review:**

This study leverages RL and PRM to optimize LLMs and proposes three aspects to for LLM enhancement: diversifying reward metrics to align LLMs to various human preference dimensions, using LLM to automate evaluation process, and utilizing working memory mechanism to incorporate past information into the reward. The idea is interesting and the proposal is well-structured, revealing a potentially effective method to improve LLMs' ability.

However, this proposal can be strengthened by further clarifying what specific tasks or LLM abilities is this method targeted on, and choosing evaluation benchmarks according to the target. If this method focuses on LLM alignment, then benchmarks such as Alignbench may be better than SQuAD and NarrativeQA.

---

### Official Review · ~Jiajun_Xu3 · 2024-11-11
**Innovative proposal and approach**

**Rating:** 9
**Confidence:** 4

**Review:**

This proposal presents an innovative approach to improving large language model (LLM) alignment by advancing reward modeling through a fine-grained, multi-dimensional framework. The authors outline a model combining reinforcement learning (RL), process reward modeling (PRM), and a memory mechanism, aimed at optimizing complex multi-step reasoning tasks beyond current capabilities.
Strengths: The proposal clearly defines the target problem, clarify the gap in current LLM alignment approaches and presents an innovative approach to address them.
Weaknesses: Some of the citations seem to lack timeliness.

---

### Official Review · ~XueZeng1 · 2024-11-11
**Innovative Solution**

**Rating:** 9
**Confidence:** 4

**Review:**

This proposal offers  detailed method to solve problems of Human Feedback (RLHF).

Strength:Firstly,It proposes an effective improvement approach by enhancing reward modeling and integrating memory mechanisms to address complex reasoning tasks.Secondly,the introduction of fine-grained reward metrics such as "contextual coherence" and "logical consistency" is a significant highlight;Third,the use of another LLM as an automated evaluator has considerable advantages. It not only effectively reduces the cost of manual labeling but also makes the evaluation process more dynamic.Finally,the addition of the working memory module and its dynamic information management method guided by the gating mechanism provide strong support for the model to conduct multi-step reasoning.

Weakness:Firstly,it may lead to a higher demand for computational resources during the actual running process.Secondly,although the use of an LLM as an automated evaluator can bring many conveniences, the LLM itself may also have certain biases. If the automated evaluator has a systematic bias, it may affect the training and evaluation results of the model, causing the model to develop in a direction that is not optima.

---

### Official Review · ~Xin_Chen65 · 2024-11-11
**Well-crafted  proposal**

**Rating:** 9
**Confidence:** 4

**Review:**

The proposal presents a novel approach to enhancing the reasoning capabilities of large language models (LLMs) through the integration of reinforcement learning (RL), process reward models (PRMs), and memory augmentation.
Strength: (1) The proposal is well-structured, with a clear background, definition, related work; (2) The proposal introduces a unique combination of RL, PRM, and memory mechanisms to address the limitations of traditional alignment methods for LLMs. This seems to be a novel and effective approach.
Weakness: (1)It would be helpful to include a section on the expected outcomes and how they will be measured; (2) It could benefit from a more detailed discussion on the potential challenges and limitations of the proposed method.

---

### Official Review · ~Mingdao_Liu1 · 2024-11-12
**Review for "Fine-Grained Reward Modeling in LLMs: An RL, PRM, and Memory-Augmented Approach for Advanced Reasoning"**

**Rating:** 10
**Confidence:** 3

**Review:**

The proposal aims to address the limitations of traditional RLHF by introducing fine-grained reward metrics and working memory. The proposal plans to use feedback from a PRM and a memory module to optimize an RL agent with PPO. Further, the proposal plans to iterate the evaluation and feedback cycle for multiple rounds, allowing the model to learn from past memory and PRM evaluations.

The proposal includes all required sections for a proposal and clearly outlines the experimental setup, the evaluation datasets, and the baselines to compare.

---

### Official Review · ~Ruilin_Hu2 · 2024-11-12
**Review of "Fine-Grained Reward Modeling in LLMs: An RL, PRM, and Memory-Augmented Approach for Advanced Reasoning"**

**Rating:** 9
**Confidence:** 5

**Review:**

The proposal presents a sophisticated approach to advancing large language model (LLM) alignment by addressing limitations in current reinforcement learning frameworks. By integrating fine-grained reward metrics (e.g., contextual coherence, logical consistency) and a memory-augmented mechanism, the authors offer a dynamic, adaptive solution to enhance multi-step reasoning tasks. This method reduces manual annotation costs through automated feedback and is validated against notable datasets. Overall, the proposal is robust, innovative

---

### Official Review · ~Kairong_Luo1 · 2024-11-12
**A good combination appoach**

**Rating:** 8
**Confidence:** 4

**Review:**

Strength:
1. The combination sounds reasonable, add intermediate reward for inter-reasoning actions, hopefully can lead to a better result;
2. The problem setting and method is clean( while math notation should improve, like $k$ and $T$ are not explained or used)
3. Complete investigation about benchmark and dataset.

Weakness:
1. Although the direction is interesting, important details are required, like how to measure "consistency" and so on. Or some possible directions or where your confidence are based on is OK;
2. The combination needs more explanation. The works seem like three sub-papers, including diverse rewards, process reward, and memory model. What is the relations between them? Why do they benefit each other? Otherwise, in my opinion, one sub-paper is hard enough task to tune.

---

### Decision · Program_Chairs · 2024-11-18

**Decision:**

Strong Accept (Long Presentation)

**Comment:**

**Fine-grained Reward Modeling in LLMs: RL, PRM, and Memory-Enhanced Methods for Advanced Reasoning**

**2.2.1 Key Innovations**
1. Diversified reward metrics
2. Introduction of an additional LLM for automated feedback evaluation
3. Implementation of a working memory mechanism

**2.2.2 Additional Key Information**
None

**2.2.3 Advantages**
1. Incorporates memory-enhancing networks such as Differentiable Neural Computers (DNC)
2. Utilizes the bAbI dataset from FAIR for reading comprehension and Q&A evaluation

**2.2.4 Areas for Improvement**
1. Incomplete definition of \( M(x) \)
2. Lack of detailed explanation on the purpose of the additional LLM for automated feedback evaluation
3. Unclear resolution of the complexity in multi-faceted rewards
4. Insufficient discussion on why RLHF no longer meets the requirements

**2.2.5 Recommendations**
1. Clearly define objectives and expected outcomes
2. Elaborate on the challenges addressed by this work